# Etiology of Molar-Incisor Hypomineralization (MIH): A Cross-Sectional Study of Saudi Children

**DOI:** 10.3390/children8060466

**Published:** 2021-06-02

**Authors:** Latifa Alhowaish, Laila Baidas, Mohammed Aldhubaiban, Lanre L. Bello, Nouf Al-Hammad

**Affiliations:** Pediatric Dentistry and Orthodontics Department, College of Dentistry, King Saud University, Riyadh 12372, Saudi Arabia; LBaidas@KSU.EDU.SA (L.B.); maldhubaiban@ksu.edu.sa (M.A.); LBello@ksu.edu.sa (L.L.B.); nalhammad@ksu.edu.sa (N.A.-H.)

**Keywords:** MIH, molar-incisor hypomineralization, children, enamel, etiology, pediatric dentistry, dental anomalies

## Abstract

(1) Background: Molar-incisor hypomineralization (MIH) is a common clinical condition with critical negative consequences for dental health. The etiology of MIH is still not completely understood, although several theories have been suggested. (2) Aim: To investigate the etiology of MIH defects in a sample of Saudi school children. (3) Method: A total of 893 school children in the age range of 8–10 years participated in the study. The sample was taken from Riyadh City, Saudi Arabia. The participating children were examined for MIH using the European Academy of Pediatric Dentistry Criteria. The children’s parents were asked about the child’s pre, peri-, and postnatal condition utilizing a structured and validated questionnaire. (4) Results: A total of 362 children (168 males and 194 females) were affected with MIH, for a prevalence of 40.5%. Among all analyzed etiological factors, only jaundice was found to be significantly associated with MIH in children (OR = 1.35, *p* = 0.047). Multivariate logistic regression analysis confirmed that the only significant etiological factor for MIH was newborn jaundice (*p* = 0.04). (5) Conclusion: Newborn jaundice was the only etiological factor that showed a significant association with MIH in the studied Saudi school children.

## 1. Introduction

In 2001, Weerheijm and his colleagues were the first to describe a developmental enamel defect, and name it molar-incisor hypomineralization (MIH) [1]. This defect is of qualitative nature, and presents as reduced mineralization and increased porosity of the enamel structure. It can affect at least one and up to four first permanent molars, with or without the involvement of the permanent incisors [1]. Enamel defects in MIH range from demarcated opacities with white or yellow/brown discoloration to a severe form of MIH that is characterized by post-eruptive enamel breakdown due to the combination of occlusal load, and an inherently weak enamel structure [1]. 

There are multiple negative consequences of MIH, including increased caries risk, breakdown, aesthetic problems, dental sensitivity, and tooth loss. These clinical problems, combined with the fact that MIH can be difficult to manage in children and young adolescents from a behavior management point of view, lead to a challenging clinical situation [2,3,4,5]. 

In a review of 70 global studies of MIH prevalence, the authors stated that the pooled prevalence of MIH in children is 14.2%, with a range from 2.8% to 44% [6]. Several dimensions of MIH have gained considerable attention from researchers. Prevalence, etiology, and management are the most studied aspects of MIH. For the management of all dental conditions, a reasonable understanding of the condition’s etiology is required [7]. The authors of a 70-study review emphasized that the etiology of MIH must be clearly identified to help reduce its occurrence [6]. 

In the Arab region, in general, and in Saudi Arabia specifically, prevalence of molar incisor hypomineralization studies are very limited. The prevalence of MIH in Jordan was found to be 17.6%. In Iraq, a study with a sample of children ranging from 7–9 years of age reported a prevalence of 18.6% [8,9]. 

In Saudi Arabia, only two prevalence studies have been published. One study was conducted in Jeddah, a city in the western part of the country, and reported an MIH prevalence of 8.6% among 8- to 12-year-old children [10]. Another study, which is the first part of the present study conducted in Riyadh, the capital of Saudi Arabia, found a much larger MIH prevalence of 40.7% in school children [11]. 

The etiology of MIH in several communities has been detailed in the literature. Although the etiology of MIH remains unclear to date, two theories have been suggested: environmental insults during the prenatal, perinatal, and postnatal periods or a genetic origin [12,13]. The genetic origin of molar incisor hypomineralization was highlighted by Vieira and Kup, and they believe that genetic variations in genes involved in enamel formation can be confirmed, and therefore, genetic etiology must be considered [13]. 

Of the MIH etiology studies, only one—a Swedish study—had a prospective design, and it did not identify a single etiological factor [14]. A very interesting umbrella review published recently analyzed three meta-analyses on the etiological factors of MIH [15]. None of the conducted analyses identified a single etiological factor that is significantly associated with MIH.

A significant number of events in the pre, peri-, and postnatal periods have been implicated in the etiology of MIH. The prenatal events include maternal medical problems, urinary tract infection in the third trimester of pregnancy, maternal anxiety, and smoking [12]. Potential etiological factors during the perinatal period include premature birth, cesarian section, difficult delivery, and hypoxia [16]. During the postnatal period, potential factors related to MIH include early childhood illness, including respiratory diseases, infections, and fever, antibiotics use, prolonged breastfeeding, and environmental pollution [16]. 

It has been suggested that any systemic physiological stress can compromise ameloblast activity at any time during enamel formation [17]. Because dental enamel is a highly specialized structure with a considerably limited regeneration ability, any disturbance during its formation may result in a clinically visible, and irreversible defect [18]. 

Due to the considerably high prevalence of MIH in Saudi children, the burden of its treatment on children, parents, and operators, and the need to prevent etiological factors from increasing the prevalence of the condition, the aim of this study was to investigate the possible etiological factors of MIH in Saudi children.

## 2. Materials and Methods

### 2.1. Setting and Study Population

This present study has a retrospective cross-sectional design. Prior to the study, ethical approval was obtained from the institutional ethics committee of the College of Dentistry at King Saud University under number (FR 0154), followed by approval from the Office of the President General for Education in Riyadh. The sample consisted of children in the age range of 8–10 years attending elementary schools representing all areas of the city of Riyadh (central, northern, southern, eastern and western). Additionally, a group of similar-aged children attending dental teaching hospitals at King Saud University was included. Information sheets detailing the study’s aim, and the nature of the clinical examination and survey questions were sent to the families of the school children prior to the clinical examination. Informed consent was obtained from parents who were willing for their children to participate in our study. Similarly, for the children recruited from the dental hospital, one of the authors (N.A.-H.) approached the parents, explained the study, and obtained informed consent from those who agreed to participate.

The inclusion criteria included the following:Children aged 8–10 yearsSaudi children who were lifelong residents of RiyadhErupted first permanent molars, and a minimum of six permanent incisors

The exclusion criteria included the following:Children with signs of fluorosis, tetracycline staining, amelogenesis imperfecta, and generalized enamel hypoplasiaUndergoing orthodontic treatmentOpacities confined to the incisors onlyAbsence of parental consent to participate

### 2.2. Examination

A pilot examination was performed by two trained pediatric dentists prior to the commencement of data collection for the study. The examiners were trained and calibrated on 20 MIH patients who were not part of the study. Intra- and inter-examiner reliability for the dental examination of MIH patients was assessed using Cohen’s kappa score. Intra-examiner reliability was 0.91 for examiner 1 (N.A.-H.) and 0.89 for examiner 2 (M.A.). The inter-examiner reliability score was 0.87.

During the examination, the first permanent molars, and incisors (the index teeth) were cleaned using a cotton roll, and the children were seated on a chair facing a light source. The children were asked to close their mouths for a short time, and then reopen it to ensure teeth are reasonably wet prior to examination. The criteria initially developed by the European Academy of Pediatric Dentistry in 2003, and revised in 2010 were followed [19,20].

### 2.3. Questionnaire

The scientific evidence was critically reviewed to identify the suspected etiological factors for MIH that have been hypothesized. Following careful review, a structured and validated questionnaire targeting the mothers of all the participants was designed (Table 1). The questionnaire was completed either in a face-to-face interview or over the phone by one of the authors (N.A.-H.). Those children whose mothers could not be reached either because she passed away or she is divorced, were excluded. The first part recorded demographic characteristics, including the child’s age, sex, place of residence and birth. The second part asked about prenatal, perinatal and postnatal conditions. Delivery method, any delivery complications, diarrhea, asthma, otitis media, jaundice, nutrition difficulties, feeding practices, respiratory problems, antibiotic intake, and any abnormalities encountered during the three periods were also collected. All collected data were tabulated and entered into a FOX PRO program. The Statistical Package for Social Sciences version 20 (SPSS) was utilized to analyze and interpret the data (SPSS, Inc., Chicago, IL, USA).

## 3. Results

A total of 893 (461 female and 432 male) 8- to 10-year-old children were examined, and their parents completed the questionnaire in full. The total number approached was 924, however we only included the 893 who completed the questionnaire by their mothers. A total of 362 children (168 males and 194 females) had MIH, for a prevalence of 40.5%. The sample (893) included 273 (30.6%) 8-year-old, 320 (35.8%) 9-year-old, and 300 (33.6%) 10-year-olds. The 362 MIH-affected children were distributed by age as follows: 105 (29%) were 8 years old, 143 (39.5) were 9 years old, and 114 (31.5) were 10 years old. When the distribution of defects was examined according to age and sex, no significant difference was found. (Table 2).

The distribution of the etiological factors in the children affected with MIH (MIH+) and not affected (MIH-), and the association of etiological factors with MIH are presented in Table 3. The chi-square test showed that the presence of jaundice shortly after birth was significantly associated with MIH (*p* = 0.04). All other investigated etiological factors were not significantly associated with MIH.

Multivariate Binary logistic regression analysis was applied. The dependent variables were MIH+ (affected) and MIH- (not affected), and explanatory independent variables (age, gender and etiological factors) were considered significant if the *p*-value < 0.05 (Table 4). According to logistic regression full model, only newborn jaundice was found to be significantly associated with MIH in children (OR = 1.35, *p* = 0.047). The goodness-of-fit of logistic regression full model confirmed by the Hosmer and Lemeshow test was not significant (*p* = 0.678), and 60.2% of the data were correctly classified. 

Table 5 showed the analysis of forward stepwise multivariate binary logistic regression, which confirmed that the jaundice is significantly associated with MIH (OR = 1.36, *p* = 0.04). The goodness of fit of the reduced model confirmed with the Hosmer and Lemeshow test was not significant (*p* = 0.1245), but the test model was significant (*p* = 0.04), and 59.5% of the data were correctly classified.

## 4. Discussion 

The findings of the present study highlight the relatively high prevalence of MIH among school children in the capital, and largest city of Saudi Arabia, Riyadh. We found a prevalence of 40% in the studied sample. In terms of international studies, this number is comparable to a prevalence study in Brazilian children, which found a prevalence of 40.2% in a sample of 7- to 13-year-olds [21]. In contrast, in a sample from Jeddah, located in western Saudi Arabia, the reported prevalence was 8.6% in 8- to 12-year-old children [10]. The wide variation between the two studies’ findings can be explained by the differences in sampling techniques. The sample size in our study was almost 3.5 times the size of the sample in the Jeddah study, and, according to Elfrink et al., a minimum of 300 subjects is required to determine MIH prevalence [22]. Additionally, their study used a convenience sample of patients attending dental clinics only, and all nationalities were included. In the current study, a random sample of children from schools in all areas of the city, and children attending a dental hospital was invited to participate. In the studied sample, no significant differences in MIH prevalence were found between sex and age groups, which is in agreement with previous similar studies [10,23,24]. 

The main aim of the present study was to identify the possible etiological association of MIH with pre, peri-, and postnatal events in Saudi school children. It is important to mention that such associations are generally difficult to identify, and conclusions should be carefully considered.

We relied mainly on mothers’ recall of any abnormalities. The risk was minimized as much as possible through structured questioning conducted by one investigator, which could help the mother recall any related factors. Additionally, the questionnaires information was only taken from the child’s mothers. Fathers or other members of the family were not questioned because they would induce higher risk of recall bias when asked about pre, peri-, and post-natal events. All suggested etiological factors that were described in previous scientific literature were considered [10,16,17,23] 

In the present sample, only one of the suggested factors show statistically significant association with MIH: newborn jaundice. Multivariate logistic regression further confirmed the positive association between MIH and jaundice in the current sample of school children from Riyadh.

Birth complications, prematurity, and cesarean section did not show a statistical association with MIH. Fatturi et al. reviewed 24 observational studies of MIH etiology. The authors concluded that maternal psychological illness, delivery complications, and cesarean section showed significant correlations with MIH [16]. However, this result must be interpreted with caution, because it included only observational studies, which have a high risk of bias and inconsistency [15]. Low birth weight neonates had a three times higher risk of MIH in a number of other observational studies, and a meta-analysis [25]. On the other hand, in a study on the association between MIH, and peripartum events in a French sample, the authors reported that there was no association between prematurity and MIH [26]. Additionally, Allazzam et al. agreed with the present findings, and reported no significant association between peri- and prenatal events and MIH [10].

Respiratory problems, asthma and early childhood illness were not statistically significant etiological factors in the occurrence of MIH in our sample of children. Silva et al. reported that early childhood disease, specifically fever, might be an offending etiological factor in MIH; however, they suggested that further prospective studies with better control of confounding factors are necessary to better understand the real etiology of MIH [12,17]. It has been explained that hypoxic conditions, which are theoretically associated with respiratory disorders, can lead to disturbances of amelogenesis [12]. Drugs, especially antibiotics, have been investigated as a possible etiological factor for MIH. In a systematic review, medications were not significantly associated with MIH, which is in agreement with our findings [27]. 

In a multivariate logistic regression model, we found a significant association between jaundice and MIH with a correlation coefficient of 0.3. This association is not considered very strong; however, it is statistically significant. Jaundice, which is also known as hyperbilirubinemia, can be detected when the total serum level of bilirubin exceeds 5 mg per dL, a condition that is not uncommon among newborns [28,29]. It is a common practice in children’s hospitals to screen the newborn for jaundice, and the treatment modality for those children with mild hyperbilirubinemia is phototherapy which was the case in our sample whose mothers were questioned specifically about treatment under light for jaundice [30]. 

The effect of newborn jaundice on dental disorders and on MIH specifically has rarely been considered in previous etiological studies. In the reviewed literature, we found only two studies that mentioned jaundice, and explicitly asked about it on a parental questionnaire [10,23]. Neither study correlated jaundice with the presence of MIH in their samples; however, it is worth mentioning that in both study groups, the prevalence of jaundice and MIH were much lower than those in our present study. To date, the etiological mechanism underlying MIH is still not understood. Both environmental changes and genetic predisposition are possible etiologies. A complete analysis of the factors that lead to MIH is crucial to improve preventive and treatment protocols for such problematic dental conditions, specifically in children.

## 5. Conclusions

In conclusion, within the limitations of the current study, which was a retrospective study with a known recall bias risk, we highlighted for the first time that jaundice is another possible perinatal etiological factor in the occurrence of MIH in the examined children. Further prospective well-designed longitudinal studies that use accurate medical documentation are needed to confirm the etiopathogenesis of molar incisor hypomineralization.

## Figures and Tables

**Table 1 children-08-00466-t001:** Questionnaire given to participants’ mothers.

The question given to mothers of participating children
Age?
Gender?
Place of residence and birth?
Was your child full term/pre-term?
Were there any birth complications?
Was the child delivery normal/Caesarean?
Did your child have asthma at or before the age of three?
Did your child have airway infection at or before the age of three?
Did your child have otitis media at or before the age of three?
Did your child have diarrhea at or before the age of three?
Was your child diagnosed with new-born jaundice requiring treatment under light?
Did your child have nutrition problems?
Was your child breastfed?
Did your child receive treatment with antibiotics at or before the age of three?
Any other medical events you would like to mention?

**Table 2 children-08-00466-t002:** MIH distribution according to age and sex.

		MIH + (362)		MIH- (531)		Total (893)		
		*n*	%	*n*	%	*n*	%	*p*-Value
Sex	Male	168	38.9	264	61.1	432	48.4	0.341
	Female	194	42.1	267	57.9	461	51.6	
Age	8 years	105	29	168	31.64	273	30.6	0.167
	9 years	143	39.5	177	33.33	320	35.8	
	10 years	114	31.5	186	35.02	300	33.6	

MIH+ = affected with MIH, MIH- = not affected with MIH.

**Table 3 children-08-00466-t003:** Distribution of etiological factors in children with and without MIH.

Etiological Factors		MIH + (362)		MIH- (531)		Total (893)		
		*n*	(%)	*n*	(%)	*n*	(%)	*p*-Value
Premature birth	yes	17	4.7	25	4.7	42	4.7	0.993
	no	345	95.3	506	95.3	851	95.3	
Childbirth complications	yes	16	4.4	20	3.8	36	4	0.626
	no	346	95.6	511	96.2	857	96	
Diarrhea	yes	49	13.5	79	14.9	128	14.3	0.574
	no	313	86.5	452	85.1	765	85.7	
Asthma	yes	68	18.8	91	17.1	159	17.8	0.528
	no	294	81.2	440	82.9	734	82.2	
Otitis media	yes	68	18.8	86	16.2	154	17.2	0.368
	no	294	81.2	445	83.8	739	82.8	
Normal birth	yes	312	86.2	457	86.1	769	86.1	0.949
	no	50	13.8	74	13.9	124	13.9	
Newborn jaundice	yes	117	32.3	138	26.0	255	28.6	0.040 *
	no	245	67.7	393	74.0	638	71.4	
Airway infection	yes	14	3.9	14	2.6	28	3.1	0.300
	no	348	96.1	517	97.4	865	96.9	
Nutrition problems	yes	3	0.8	2	0.4	5	0.6	0.374
	no	359	99.2	529	99.6	888	99.4	
Breast feeding	yes	297	82.0	442	83.2	739	82.8	0.643
	no	65	18.0	89	16.8	154	17.2	
Antibiotics use	yes	205	56.8	299	82.8	504	56.5	0.107
	no	97	26.9	167	46.3	264	29.6	
	I don’t know	60	16.6	64	17.7	124	13.9	

MIH+ = affected with MIH, MIH- = not affected with MIH, * indicates statistical significance.

**Table 4 children-08-00466-t004:** Multivariate binary logistic regression of etiological factors for MIH.

Multivariate Binary Logistic Regression Analysis (Full Model)					Goodness of Fit of the Model		
Explainatory Variable	B	OR	*p*-Value	95% CI	Tests of Model Coefficients	Hosmer and Lemeshow Test	Percentage Correct Classification
Age	−0.028	0.973	0.748	0.748–1.152	0.798	0.678	60.2%
Gender	0.146	1.157	0.299	0.299–1.523			
Premature birth	−0.042	0.959	0.899	0.497–1.848			
Birth complications	0.088	1.092	0.802	0.548–2.175			
Diarrhea	−0.175	0.840	0.391	0.564–1.251			
Asthma	0.127	1.136	0.491	0.790–1.631			
Otitis media	0.160	1.173	0.396	0.811–1.697			
Normal birth	−0.037	0.963	0.854	0.648–1.433			
Jaundice	0.300	1.349	0.047 *	0.999–1.822			
Nutritional problems	0.621	1.861	0.504	0.301–11.499			
Airway infection	0.347	1.415	0.376	0.656–3.054			
Brest feeding	0.212	1.236	0.800	0.241–6.345			
Antibiotics use	−0.048	0.953	0.743	0.717–1.268			
Constant	−0.280	0.756	0.727				

* indicates statistical significance.

**Table 5 children-08-00466-t005:** Stepwise (Wald) Multivariate Binary logistic regression of etiological factors for MIH.

Forward Stepwise (Wald) Multivariate Binary Logistic Regression Equation (Reduced Model)					Model Goodness of Fit		
Explainatory Variable	B	OR	*p*-Value	95% CI	Tests of Model Coefficients	Hosmer and Lemeshow Test	Percentage Correct Classification
Newborn Jaundice	0.307	1.360	0.040 *	1.014–1.824	0.04	0.124	59.50%
Constant	−0.473	0.623	0.000				

* indicates statistical significance.

## Data Availability

Data supporting the findings of the present study can be requested from authors.

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
