# Peer review of "Etiology of Molar-Incisor Hypomineralization (MIH): A Cross-Sectional Study of Saudi Children"

_children, 2021, doi:10.3390/children8060466_

Round 1
Reviewer 1 Report
Thank you for resubmit the revised manuscript.
The authors generally responded to my comments appropriately but there is still serious problems in statistics.
I think that the statistical analysis was not conducted correctly.
For example, what was the explanatory variable in stepwise multivariate logistic regression?
This problem in statistical analysis is very important.
Therefore, I hope that the authors reanalyze the data.
Thank you.
Author Response
Many thanks for your valuable comments.
We have followed your suggestion and re-analyzed the data for a second time and included further details to facilitate analysis understanding by the readers.
Kindly, see below the re-analyzed results with the explanatory variable explained as well.
Multivariate Binary logistic regression analysis was applied. The dependent variables were MIH+ (affected) and MIH- (not affected), and explanatory independent variables (age, gender, and etiological factors) were considered significant if the P-value <0.05 (Table 3). According to logistic regression full model, only jaundice was found to be significantly associated with MIH in children (OR=1.35, P=0.047). The goodness-of-fit of logistic regression full model confirmed by the Hosmer and Lemeshow test was not significant (P=0.678), and 60.2 % of the data were correctly classified.
Table 4. Stepwise Multivariate logistic regression of etiological factors for MIH.
|
Multivariate Binary logistic regression analysis (full model) |
Goodness of fit of the model |
||||||
|
Explainatory Variable |
B |
OR |
P-value |
95% CI |
Tests of model Coefficients |
Hosmer and Lemeshow test |
Percentage Correct classification |
|
Age |
-0.028 |
0.973 |
0.748 |
0.748-1.152 |
0.798 |
0.678 |
60.2% |
|
Gender |
0.146 |
1.157 |
0.299 |
0.299-1.523 |
|||
|
Premature birth |
-0.042 |
0.959 |
0.899 |
0.497-1.848 |
|||
|
Birth complications |
0.088 |
1.092 |
0.802 |
0.548-2.175 |
|||
|
Diarrhea |
-0.175 |
0.840 |
0.391 |
0.564-1.251 |
|||
|
Asthma |
0.127 |
1.136 |
0.491 |
0.790-1.631 |
|||
|
Otitis media |
0.160 |
1.173 |
0.396 |
0.811-1.697 |
|||
|
Normal birth |
-0.037 |
0.963 |
0.854 |
0.648-1.433 |
|||
|
Jaundice |
0.300 |
1.349 |
0.047* |
0.999-1.822 |
|||
|
Nutritional problems |
0.621 |
1.861 |
0.504 |
0.301-11.499 |
|||
|
Airway infection |
0.347 |
1.415 |
0.376 |
0.656-3.054 |
|||
|
Brest feeding |
0.212 |
1.236 |
0.800 |
0.241-6.345 |
|||
|
Antibiotics use |
-0.048 |
0.953 |
0.743 |
0.717-1.268 |
|||
|
Constant |
-0.280 |
0.756 |
0.727 |
|
|||
*indicates statistical significance.
Table 5. showed the analysis of forward stepwise multivariate Binary logistic regression, which confirmed that the jaundice is significantly associated with MIH (OR=1.36, P=0.04). This model has the goodness of fit with the Hosmer and Lemeshow test that was not significant (P=0.1245), and the test model was significant (P=0.04), and 59.5 % of the data were correctly classified.
Table 5. Stepwise (Wald) Multivariate Binary logistic regression of etiological factors for MIH.
|
Forward Stepwise (Wald) Multivariate Binary logistic regression Equation (reduced Model) |
Model Goodness of fit |
||||||
|
Explainatory Variable |
B |
OR |
P-value |
95% CI |
Tests of model Coefficients |
Hosmer and Lemeshow test |
Percentage Correct classification |
|
Jaundice |
0.307 |
1.360 |
0.040* |
1.014-1.824 |
0.04 |
0.124 |
59.50% |
|
Constant |
-0.473 |
0.623 |
0.000 |
|
|||
*indicates statistical significance.

Reviewer 2 Report
Thank you for answering all my questions and for having modified the text as suggested.
Best regards
Author Response
Many thanks for your review and wishes.
Kind regards.,
Reviewer 3 Report
The article has been improved.
Author Response
Many thanks for your kind review.
Best regards.,
Round 2
Reviewer 1 Report
Thank you for submit the revised manuscript.
The authors provided the significant data.
This manuscript is a resubmission of an earlier submission. The following is a list of the peer review reports and author responses from that submission.
Round 1
Reviewer 1 Report
Thank you for providing us the opportunity to review your manuscript. I understand that your study aimed to investigate the cause of molar incisor hypomineralization. In children, hypomineralization leads to the enamel structure defects, aesthetic disability, severe dentin hypersensitivity, and occlusal collapse. Therefore, I think that it is very important to investigate the causes of hypomineralization. However, there are some major concerns and shortcomings in the content that I wish to point out.
1. Molar incisor hypomineralization is currently considered as one of the factors associated with a genetic component. Your study did not include the aspect of genetic I believe that you should discuss this point in the manuscript. Please refer to the following reference: Vieira A.R. and Kup E. (2016). On the Etiology of Molar-Incisor Hypomineralization.
2. P3 L101. I think it is necessary for you to analyze the site of the hypomineralization, the first molar or incisor, and the degree of molar incisor hypomineralization. Is it not sufficient for you to examine if molar incisor hypomineralization existed?
3. P3L113. I think that it is very difficult for parents, especially mothers, to answer these questions accurately. Perinatal period is a period when the baby is in the mother’s womb, and the duration of the perinatal period is quite long, from 22 weeks to 42 weeks. Whereas, the prenatal period is from fertilization to 21 weeks, and the postnatal period starts from 7 days of childbirth. I think the parents would not accurately remember the events from these periods because they had happened about 10 years previously.
4. As per my understanding, neonatal jaundice has many different types. You may refer to the following paper: https://www.ncbi.nlm.nih.gov/pmc/articles/PMC5408871/pdf/MFP-11-16.pdf
5. P3 L127. This study included 893 participants. What was the questionnaire collection rate?
6. P4 L148–149. The authors should show the goodness-of-fit model of stepwise logistic regression. There is no description about the statistical analysis methods.
7. P5 L171. You mention “The risk was minimized as much as possible through structured questioning conducted by one investigator.” You may show the “structured questioning.” It will dramatically increase the reproducibility of this study.
8. Table 2. I think that there are many participants who answered “I don’t know” to the question regarding antibiotics use. The authors may contact a medical institution to find out the antibiotics use.
Reviewer 2 Report
The present study aimed to identify the possible etiological association of MIH with pre, peri- and postnatal events in a sample of Saudi school children. This research appears well performed and leads to significant results.
Introduction.
- The sentence "There are multiple negative consequences of MIH, including increased caries risk, breakdown, esthetic problems, dental sensitivity and tooth loss" needs adequate references. I could suggest to add doi:10.3390/app11041823.
- The sentence "In a review of 70 global studies of MIH prevalence..." has two references (#5 and 6) but it seems that the mentioned study is referred only to reference 5. Please adjust.
- "In the Arab region in general and in Saudi Arabia specifically, prevalence studies are very limited." Prevalence of what? Please specify.
- Why do authors think that so different values of MIH prevalence (8.6% vs 40.7%) have been found in similar sample subjects? Please move this part and discuss this aspect in the Discussion section.
Materials & Methods.
- Please provide the ethical approval number.
- "The children were asked to close their mouths for a short time and then reopen it." Please explain the reason for this.
- Was the questionnaire performed directly with the subject enrolled in the study or with his/her parents?
Discussion.
Only at the end of the article, authors reveal the retrospective nature of their study. Since this is a fundamental aspect of the article, I would expect authors to specify the study retrospective nature through all the text: Title, Abstract and Materials and Methods. Moreover and consequently, the period of enrollment has to be specified. It has to be stated also in which manner the parents have been recalled to answer the questionnaire and how many patients/parents were excluded due to the impossibility to reach them.
Reviewer 3 Report
Although the topic is already widely discussed in the literature, the association between jaundice and MIH is interesting. I would ask the Authors to enter the protocol number of the Ethics Committee and also a figure showing the questionnaire with all the questions administered.